# Teachers’ Emotional Intelligence, Burnout, Work Engagement, and Self-Efficacy during COVID-19 Lockdown

**DOI:** 10.3390/bs13040296

**Published:** 2023-03-30

**Authors:** Alessandro Geraci, Laura Di Domenico, Cristiano Inguglia, Antonella D’Amico

**Affiliations:** Department of Psychology, Educational Science and Human Movement, University of Palermo, Viale delle Scienze Edificio 15, 90128 Palermo, Italy; laura.didomenico@community.unipa.it (L.D.D.); cristiano.inguglia@unipa.it (C.I.); antonella.damico@unipa.it (A.D.)

**Keywords:** emotional intelligence, ability EI, self-reported EI, burnout, work engagement, self-efficacy, teacher, remote teaching, school, COVID-19

## Abstract

Teachers’ psychological well-being is a crucial aspect that influences learning in a classroom climate. The aim of the study was to investigate teachers’ emotional intelligence, burnout, work engagement, and self-efficacy in times of remote teaching during COVID-19 lockdown. A sample of 65 teachers (M_age_ = 50.49), from early childhood through lower secondary education, were recruited during a period of school closure to answer self-report questionnaires and other measures assessing study variables. Results showed that during the COVID-19 pandemic, teachers reported higher levels of burnout and lower levels of self-esteem due to multiple challenges related to remote teaching and the growing sense of insecurity regarding health safety in the school environment. However, the negative effects of COVID-19 on teachers’ self-efficacy, work engagement, and burnout varied according to their own levels of emotional intelligence. These results demonstrate that emotional intelligence may support teachers in facing these challenges.

## 1. Introduction

The COVID-19 pandemic contributed to an unprecedented and unplanned change in the education systems around the world. Isolation, social distancing, the generalized feeling of uncertainty, and the absence of unanimously established guidelines or inequalities emerged, and the fear of contagion generated a high prevalence of anxiety, depression, stress, and sleeping disorders on the general population [1,2,3].

These effects were particularly evident in teachers since during the global health emergency, school buildings closed to protect the public’s health, and many teachers, consequently, settled for innovative ways of teaching [4,5]. Therefore, instead of traditional teaching, teachers learned an emergency remote teaching (ERT) technique defined as a transitory change from instructional delivery to an alternate mode of teaching due to crisis circumstances [6]. Consequently, teachers have experienced an important modification in their work format that has been perceived as an abrupt and unplanned change [7]. Although some teachers were ready to face the adversities, most of them had to adapt in a brief time without training and preparation [8].

However, even when school attendance was reinstated, teachers had to prevent the spread of the virus and face selective blocking and restriction measures during teaching activities [9]. However, some of the measures imposed to prevent contagion in schools also had a direct impact on the way teaching was carried out, with many classes adopting the hybrid format—with half of the students at home and half in the classroom—or bimodal [10,11], mirrored classrooms [12], small bubble groups [13], and even the use of the internet for all the school year [14]. Consequently, before the pandemic, teachers had little previous experience with integrating technology in their daily curricula [15]. This was made even more difficult by schools’ lack of digital infrastructures and equipment, together with poor digital competences of the students, especially in relation to online safety [16].

### 1.1. Teachers’ Burnout, Work Engagement, and Self-Efficacy during the Pandemic

Considering all these evidence, it is not surprising that during the pandemic period teachers tended to report higher levels of psychological difficulties, such as stress, anxiety, and depression [4,17,18,19,20,21]. In particular, the pandemic negatively affected teachers’ experience of teaching and work-life balance in terms of higher levels of burnout and lower levels of both work engagement and self-efficacy. Burnout, work engagement, and self-efficacy in teaching, indeed, are highly interconnected. Burnout is conceived to be a state in which a person perceives having no more energy and having reached a point of no return in terms of mental, emotional, and physical resources [22,23,24]. During the pandemic, studies have shown increasing levels of burnout in teachers [4,25,26,27,28]. Over time, teachers have become more exhausted and under pressure, especially because of the change from classical teaching methods into remote teaching [29]. Furthermore, studies show that female teachers, compared to men, manifest a greater vulnerability in terms of experiences and consequences of burnout [30,31]. Teachers’ burnout can affect health and well-being, increasing the chances of suffering from physical and psychological pathologies [32]. More generally, the consequences of teachers’ burnout are low satisfaction, absenteeism, substitution, early retirement, and a deterioration in instructional self-efficacy and teaching effectiveness [26,28,33]. In addition, teachers’ burnout is associated with a low teacher–student relationship, students’ poor academic success, and behavioral problems [34,35,36].

Opposite to burnout, work engagement is a persistent, positive, and satisfying work-related mental state, characterized by vigor, dedication, and absorption during work activities [37,38]. Studies show that teachers’ work engagement is associated with high satisfaction and wellbeing [39], effective instructional performance [40], and high student engagement [41] and academic success [42]. Recent studies showed that, during the pandemic, teachers’ work engagement has been negatively affected [43,44].

Lastly, self-efficacy in teaching is a particular aspect of the general construct of self-efficacy, defined in Bandura’s social-cognitive theory as a person’s belief in their ability to succeed in a particular situation [45]. Teachers’ self-efficacy corresponds to beliefs in personal and instructional ability to successfully cope with instructional tasks, obligations, and challenges [46]. In addition, it refers also to teachers’ beliefs in their ability to motivate students’ engagement and learning [8]. Past research has consistently shown the key role of teachers’ self-efficacy in fostering students’ motivation, learning, and academic achievements [47,48]. However, teachers’ self-efficacy may change over time due to several factors, such as burnout [28]. Moreover, the pandemic also affected teachers’ self-efficacy as demonstrated in the study [26] that showed how, during the pandemic, teachers manifested a lower level of instructional competence than pre-pandemic teachers. Furthermore, teachers who were teaching in a virtual classroom had the lowest efficacy scores compared to teachers teaching in a hybrid or in-person classroom [49].

### 1.2. Teachers’ Emotional Intelligence as a Protective Factor

In many recent studies, scientific research has focused on teachers’ emotional intelligence (EI) since teaching is a high-content emotional profession [50], and EI may be a crucial resource against burnout and for promoting work engagement and teaching self-efficacy. Mayer and Salovey [51] conceived EI as the ability to: (a) perceive, value, and express emotions accurately; (b) access and generate feelings that facilitate thinking; (c) understand emotions and have emotional awareness; and (d) regulate emotions and promote emotional and intellectual growth. In addition, they [51] stated that, as a form of intelligence, EI should be measured with a maximum performance test [52]. Another main theoretical model, namely trait EI, conceives EI as emotion-related dispositions and self-perceptions, which are measured using self-report [53]. However, self-report scales have also been designed on the ability EI model, which excludes traits or competencies related to emotions and shows weak association with personality dimensions [54,55]. Contrary to a maximum performance test, which measures actual abilities, ability EI self-report scales measure individuals’ perceptions about their own emotional abilities, namely self-reported EI or perceived EI (PEI) [55]. Performance-based and self-report-based measures of EI show a low convergence validity, suggesting that these two different measurement methods “are most likely tapping into different mental processes” [56] (p. 784). In addition, self-report scores might be susceptible to falsification or social desirability as well as the low accuracy of people’s perceptions about their own emotional abilities [56,57].

Regardless of the theoretical model, studies have found that EI results, both when measured using performance and self-report scales, were significantly associated with several aspects of psychological health, such as a positive mood [58], satisfaction with life [59], well-being [57], and psychological stress [60]. In addition, EI is a psychological factor that might help to buffer the effect of psychological distress [61]. Before the pandemic, some studies already demonstrated that PEI and EI are positively associated with work engagement, self-efficacy, and negatively related to burnout [55,62,63]. However, from a recent systematic review [63], it became evident that most studies assessed the relationship between teachers’ PEI and burnout while only one study used MSCEIT to measure EI. Similarly, four studies performed during the pandemic [27,64,65,66] demonstrated that teachers with higher levels of PEI presented lower levels of burnout. However, to the best of our knowledge, no studies investigated the role of teachers’ ability EI in facing adversities during the COVID-19 pandemic. Therefore, this study represents the first one to employ both performance and self-report measures to assess PEI and EI.

### 1.3. The Present Study

The purpose of this study is twofold: First, it investigates the changes from pre- to post-COVID-19 outbreak in the levels of burnout, work engagement, and self-efficacy reported by teachers. Hereby, it is hypothesized that due to the circumstances of the pandemic, burnout levels would increase whereas work engagement and self-efficacy would decrease from pre- (time 1) to post-COVID-19 (time 2). Second, it analyzes whether the changes are affected by teachers’ levels of self-reported EI abilities (PEI) and actual emotional intelligence abilities (EI). Hence, it is hypothesized that burnout, work engagement, and self-efficacy changes from time 1 to time 2 vary with the levels of PEI and EI.

## 2. Materials and Methods

### 2.1. Participants and Procedure

The study sample was composed by 65 teachers (M = 4, F = 61) with an average age of 50,49 years (SD = 8.99; age range = 29–72 years). The majority of participants worked at a school (60%), the rest was equally distributed between kindergarten (20%) and lower secondary school (20%). Most teachers worked as curricular (81.5%) whereas the rest of them worked for students with special needs (18.5%). Their professional experience ranged from 0 to 50 years (M = 19.05, SD = 11.47). At the time the research was conducted, the number of weeks that teachers spent using remote teaching ranged from 0 to 24 weeks (M = 12.58, SD = 5.35).

Teachers were recruited during their participation in the MetaEmotions at School program, an experiential training for promoting and developing EI abilities at schools [67]. Schools’ principals and teachers were informed about the research and asked to participate providing them with informed consent. In order to assess whether the COVID-19 pandemic restrictions had affected teachers’ burnout, work engagement, and self-efficacy, they were administered with a research protocol that presented two forms for some instrument (i.e., burnout, work engagement, self-efficacy): The first one asked participants to think about their work situation before the COVID-19 pandemic and was defined as T1 form, whereas the second one asked them to think about their actual work situation (i.e., during COVID-19 pandemic restrictions) and was defined as T2 form. As a result, for each scale and subscale, two scores were computed and analyzed as two distinct moments in time.

### 2.2. Instruments

#### 2.2.1. The Copenhagen Burnout Inventory—CBI

The Copenhagen Burnout Inventory [68] is a self-report measure that assesses burnout syndrome in specific domains and life contexts, including the work domain and, specifically, the school context: personal burnout (PB), work-related burnout (WB), and student-related burnout (SB). The scale consists of 19 items scored on a 5-point Likert type scale (1 = never to 5 = always). All the scales showed good internal consistency both for form A (PB: α = 0.75; WB: α = 0.84; SB: α = 0.71) and form B (PB: α = 0.86; WB: α = 0.91; SB: α = 0.84).

#### 2.2.2. The Utrecht Work Engagement Scale—UWES

The Utrecht work engagement scale [38] is a self-report tool that measures three dimensions of work engagement: vigor (VI), dedication (DE), and absorption. It is composed by nine items, in which responses are given on a frequency scale varying from 0 (never) to 6 (always). For the study, all the subscales showed good internal consistency both for form A (VI: α = 0.70; DE: α = 0.82; AB: α = 0.79) and form B (VI: α = 0.83; DE: α = 0.89; AB: α = 0.88).

#### 2.2.3. The Norwegian Teacher’s Self-Efficacy Scale—NTSES

The Norwegian Teacher’s self-efficacy scale [69] is a self-report tool that measures six aspects of teachers’ self-efficacy: instruction, adapting instruction to individual needs, cooperating with colleagues and parents, coping with change, motivating students, and maintaining discipline. This scale consists of twenty-four items, with a response scale ranging from 1 (not certain at all) to 7 (absolutely certain). All the scales showed good internal consistency both for the form A (instruction: α = 0.93; adapting instruction to individual needs: α = 0.93; cooperating with colleagues: α = 0.89; coping with change: α = 0.85; motivating students: α = 0.87; maintaining discipline α = 0.92) and form B (instruction: α = 0.92; adapting instruction to individual needs: α = 0.91; cooperating with colleagues: α = 0.87; coping with change: α = 83; motivating students: α = 0.88; maintaining discipline α = 0.91).

#### 2.2.4. The Wong and Law Emotional Intelligence Scale—WLEIS

The Wong and Law emotional intelligence scale [70] is a self-report measure of EI. It consists of 16 items, which are scored on a 7-point Likert-type scale (1 = totally disagree to 7 = totally agree) that aims to assess people’s perception about their own emotional abilities (self-reported EI): self-emotion appraisal (SEA), others’ emotion appraisal (OEA), use of emotion (UOE), and regulation of emotion (ROE). The overall scale showed good internal consistency (α = 0.88).

#### 2.2.5. The Mayer-Salovey-Caruso Emotional Intelligence Test—MSCEIT

The Mayer-Salovey-Caruso emotional intelligence test [71] is a maximum performance ability-based measure of EI, which assesses the four branches [50]. The test is composed by 141 items divided in eight sections (tasks) that, in pairs, evaluate the four main emotional intelligence abilities: perceiving, facilitating, understanding, and managing emotions. The participants’ answers are corrected using the general consensus scores. The test allows us to obtain fifteen main scores: eight subscale scores, the four-branches scores, two area scores (experiential and strategic), and an overall score (EIQ). For the study, the overall test showed good internal consistency (α = 0.82).

#### 2.2.6. The Remote Teaching Questionnaire

The remote teaching questionnaire was designed ad hoc and used in this study to gather information about the extent to which teachers used remote teaching and how it affected their work. The questionnaire is composed of seven items scored on a 4-point scale ranging from 1 (not at all) to 4 (very). Each item explored different aspects of the perceived quality of teachers’ work, space, or equipment to carry out remote teaching.

## 3. Results

### 3.1. Teachers’ Perceptions about Remote Teaching

Teachers’ answers to the remote teaching questionnaire were analyzed by computing percent frequency distribution for each item that was reported in the bar charts (Table 1). Results showed that 64.6% of teachers believed that remote teaching has affected the quality of their work whereas only 9.2% of teachers thought otherwise. In addition, 78.5% of the teachers sampled perceived their workload to be increasing due to remote teaching, and 72.3% of teachers felt that their work–life balance was negatively affected by remote teaching. Nevertheless, teachers reported to have enough adequate space at home (83.1%), proper equipment at home (84.6%), and technical skills to conduct remote teaching classes (75.3%). However, 52.3% of teachers found it difficult to conduct remote teaching classes as they had to deal with their family members.

### 3.2. Repeated Measures ANOVA and ANCOVA Results

To evaluate if teachers’ work engagement, burnout, and self-efficacy have changed due to the COVID-19 pandemic, three repeated ANOVA measurements were conducted on the scores obtained in T1 and T2 forms. For each scale, the ANOVA included the different subscale scores. Moreover, six ANCOVA measurements were performed with the aim of evaluating whether PEI and/or EI influenced the effects of the COVID-19 pandemic in work engagement, burnout, and self-efficacy: Three ANCOVA measurements included the total score of WLEIS, and the three ANCOVA results were entered as the total score of MSCEIT as covariates. Mean, standard deviation, and repeated ANOVA measurements and ANCOVA results for each variable are presented separately in Table 2, Table 3 and Table 4. Results showed that burnout mean levels (Table 2) greatly increased during the COVID-19 pandemic in each domain: personal burnout (F(1, 64) = 79.04, *p* < 0.001, η_p_^2^ = 0.55), work-related burnout (F(1, 64) = 49.89, *p* < 0.001, ηp2 = 0.44), and student-related burnout (F(1, 64) = 48.40, *p* < 0.001, η_p_^2^ = 0.43). However, when both total scores of MSCEIT and WLEIS were entered as a covariate, such differences were no longer significant. Results for work engagement (Table 3) showed that after the COVID-19 pandemic, teachers’ mean levels decreased largely for the dimensions of vigor (F(1, 64) = 40.87, *p* < 0.001, η_p_^2^ = 0.39) and dedication (F(1, 64) = 14.14, *p* < 0.001, η_p_^2^ = 0.18). Once again, repeated measurement for ANCOVA results showed that such differences vary as a function of MSCEIT and WLEIS. Finally, results for self-efficacy (Table 4) showed a decreasing mean level after the COVID-19 pandemic for the dimension of instruction (F(1, 64) = 7.78, *p* < 0.01, η_p_^2^ = 0.11), adapting instruction to individual needs (F(1, 64) = 8.55, *p* < 0.01, η_p_^2^ = 0.12), coping with change (F(1, 64) = 9.71, *p* < 0.01, η_p_^2^ = 0.13), motivating students (F(1, 64) = 8.71, *p* < 0.01, η_p_^2^ = 0.12), and maintaining discipline (F(1, 64) = 6.08, *p* < 0.05, η_p_^2^ = 0.09). However, ANCOVA results showed such differences ceased to be significant when considered MSCEIT and WLEIS.

## 4. Discussion

During these past two years, the pandemic has impacted people’s health and well-being both in a professional context and in everyday life. This study aimed to investigate whether teachers’ burnout, work engagement, and self-efficacy have shifted due to COVID-19 restrictions. In addition, it aimed to investigate whether such changes had a lower impact in people who showed higher levels of PEI and EI.

### 4.1. Effects of Remote Teaching and COVID-19 Lockdown

With regard to the first aim of the study and hypothesis, data confirmed our predictions. In line with other studies that showed that the pandemic has negatively impacted teachers’ burnout, work engagement and self-efficacy [4,25,26,27,28,49], during the pandemic, our participants reported higher levels of burnout and lower levels of work engagement and self-efficacy than before the pandemic. Moreover, consistently with other studies showing the negative impact of remote teaching [29], our participants reported that remote teaching has affected the quality of their teaching as the workload increased. Thus, it is possible that being forced to resort to remote teaching and to change their ways of teaching would have been an additional source of stress along with the ongoing fear of contagion. Even though most of the teachers had technical and digital skills to conduct the remote activities, they reported a decrease in their instructional efficacy. They had difficulties managing the classroom and the discipline, adapting activities to individual needs, and motivating the students.

### 4.2. The Role of PEI and EI

With regard to the second aim of the study and hypothesis, our findings also confirmed the initial hypotheses about the protective role of EI against burnout and for maintaining high levels of teachers’ self-efficacy and work engagement consistently with studies before [55,62,63] and during the pandemic [27,64,65,66]. Using both assessment methods focused on PEI and EI provided crucial information since we had the opportunity to confirm that both the self-perception of EI and the actual abilities in performing emotional tasks are factors that influence the way teachers react to stressful situations, such as the outbreak of COVID-19. Emotions convey key information that point out which aspect of the internal and/or external world requires immediate attention. Ignoring emotions, or not processing them properly, implicates that one cannot address properly an adverse situation. High EI individuals should process such information and orient efficiently their thoughts or behaviors to reach the desired outcome.

### 4.3. Limitations

The current study has potential limitations that must be taken into consideration. First, at the time the research was conducted, the Sicilian region was under a local lockdown due to a high rate of contagion, and consequently, schools were closed. Therefore, teachers’ perceptions might have been affected by the moment they were living in as well as remote teaching. Second, though the results are consistent with previous research, the small sample size is yet a limit. Third, the sample of the present study is almost entirely composed of females, and this could have affected the results; by contrast, the sample distribution reflects the prevalence of females in the general teaching population. Last, we are aware that considering teachers’ perceptions before and during the pandemic as a repeated measure was a methodological artifact since data were collected simultaneously. Hence, teachers’ perceptions might have not been accurate or influenced by other factors not measured in the present study.

## 5. Conclusions

The COVID-19 pandemic has deeply changed several aspects of people’s lives since they have been faced with increased instability in crucial areas of their lives, which has created ongoing pressure. Many people have been afraid for their health, while others have lost the means to provide for themselves or for their families. Like many, teachers have been affected by these problems, and additionally, they faced the pressure of having to function within an educational system that was not designed for remote interactions. All of this contributed to a complex and heavy situation for teachers, which added up to a pre-existing difficult job context. This study provides evidence that, during the pandemic, our participants reported higher levels of burnout and lower levels of work engagement and self-efficacy and that these variables are influenced by teachers’ individual differences in PEI and EI. Despite the limitations, this study contributes to the advancing research and practice related to the role of emotions in teaching, and to highlight the need for teachers to improve and develop their emotional skills to face ongoing demands and new challenges. This suggests that EI abilities are critical resources for adapting to unexpected and challenging situations and that they may become “critical tools” for teachers to adapt to the requests of today’s schools and society and to keep up with the ongoing changes. Adequate emotional processing and managing can support teachers’ ability to address changing and stressful work environments. Teachers’ stress may impair their personal health, as well as compromise their teaching efficacy. In this perspective, it is important to invest in social and emotional learning and training specifically addressed to teachers, both for improving teachers’ emotional abilities and as a medium for improving class climate and students’ emotional well-being.

## Figures and Tables

**Table 1 behavsci-13-00296-t001:** Percent frequency distribution of teachers’ answers to the remote teaching questionnaire.

Item	Not at All	Slightly	Moderately	Very
1. Do you believe that remote teaching has negatively affected the quality of your work?	9.2	26.2	49.2	15.4
2. Do you perceive an increase in your workload as a result of remote teaching?	9.2	12.3	32.3	46.2
3. Has remote teaching negatively affected your daily planning and work–life balance?	7.7	20	33.8	38.5
4. Do you have adequate space at home to conduct your remote teaching classes?	0	16.9	50.8	32.3
5. Do you have proper equipment at home to conduct your remote teaching classes?	1.6	13.8	63.1	21.5
6. Do you possess the technical skills required to manage your remote teaching classes?	1.5	23.2	61.5	13.8
7. Is remote teaching made difficult by the need to mind your children or other family members?	23.1	24.6	32.3	20

**Table 2 behavsci-13-00296-t002:** Repeated measures ANOVA and ANCOVA results for burnout.

			Descriptive Statistics		ANOVA			ANCOVA (MSCEIT)			ANCOVA (WLEIS)		
Scale	Subscale	Time	Mean	SD	F_(1,64)_	*p*	η_p_^2^	F_(1,63)_	*p*	η_p_^2^	F_(1,63)_	*p*	η_p_^2^
CB	PB	T_1_	28.72	14.99	79.04 ***	0.000	0.55	1.68	0.199	0.03	0.02	0.880	0.00
		T_2_	49.87	20.18									
	WB	T_1_	19.45	14.06	49.89 ***	0.000	0.44	1.77	0.188	0.03	0.00	0.983	0.00
		T_2_	38.13	22.30									
	SB	T_1_	19.81	14.15	48.40 ***	0.000	0.43	.35	0.557	0.01	0.06	0.814	0.00
		T_2_	35.19	22.29									

*** *p* < 0.001.

**Table 3 behavsci-13-00296-t003:** Repeated measures ANOVA and ANCOVA results for work engagement.

			Descriptive Statistics		ANOVA			ANCOVA (MSCEIT)			ANCOVA (WLEIS)		
Scale	Subscale	Time	Mean	SD	F_(1,64)_	*p*	η_p_^2^	F_(1,63)_	*p*	η_p_^2^	F_(1,63)_	*p*	η_p_^2^
UWES	VI	T_1_	5.64	0.63	40.87 ***	0.000	0.39	1.68	0.200	0.03	0.14	0.707	0.00
		T_2_	5.08	1.00									
	DE	T_1_	5.78	0.59	14.14 ***	0.000	0.18	1.83	0.181	0.03	0.05	0.817	0.00
		T_2_	5.41	0.87									
	AB	T_1_	5.64	0.65	3.35	0.072	0.05	0.25	0.620	0.00	0.14	0.706	0.00
		T_2_	5.47	0.77									

*** *p* < 0.001.

**Table 4 behavsci-13-00296-t004:** Repeated measures ANOVA and ANCOVA results for self-efficacy.

			Descriptive Statistics		ANOVA			ANCOVA (MSCEIT)			ANCOVA (WLEIS)		
Scale	Subscale	Time	Mean	SD	F_(1,64)_	*p*	η_p_^2^	F_(1,63)_	*p*	η_p_^2^	F_(1,63)_	*p*	η_p_^2^
NTSES	Instruction	T_1_	6.02	1.07	7.78 **	0.007	0.11	0.11	0.737	0.00	1.57	0.215	0.02
		T_2_	5.76	1.09									
	Adapting instruction to individual needs	T_1_	6.01	1.06	8.55 **	0.005	0.12	0.03	0.875	0.00	0.84	0.363	0.01
		T_2_	5.69	1.09									
	Cooperating with colleagues	T_1_	6.04	0.99	3.78	0.071	0.05	0.81	0.373	0.01	1.53	0.221	0.02
		T_2_	5.85	1.02									
	Coping with change	T_1_	5.67	1.11	9.71 **	0.003	0.13	0.16	0.690	0.00	1.54	0.219	0.02
		T_2_	5.40	1.06									
	Motivating students	T_1_	5.75	1.01	8.71 **	0.004	0.12	0.02	0.882	0.00	0.54	0.464	0.01
		T_2_	5.46	1.04									
	Maintaining discipline	T_1_	5.89	1.08	6.08 *	0.016	0.09	1.99	0.163	0.03	0.07	0.794	0.00
		T_2_	5.65	1.14									

** *p* < 0.01; * *p* < 0.05.

## Data Availability

Data will be made available by the authors upon request.

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
