# Peer review of "Teachers’ Emotional Intelligence, Burnout, Work Engagement, and Self-Efficacy during COVID-19 Lockdown"

_behavsci, 2023, doi:10.3390/bs13040296_

Round 1

Reviewer 1 Report

It is suggested to add in lines 39 to 52. 

For https://www.researchgate.net/publication/361593888_La_mediacion_didactica_del_profesorado_italiano_de_ELE_ante_el_desafio_del_confinamiento_domiciliario_por_COVID-19, factors such as the lack of digital infrastructures prior to confinement, the absence of training and safety plans for schoolchildren's navigation are identified as sources of the feeling of pedagogical burnout.

These feelings are not only confined to the Italian context, but also to teachers of similar studies in the European context, as pointed out by https://revistas.uned.es/index.php/REEC/article/view/29017.

The theoretical framework is linear and an adequate correlation is established between the causes and possible effects of confinement to the detriment of teachers' emotional education. Furthermore, although it is not the central theme of the study, it is also necessary to correspond to a generalised feeling that is extrapolated to both families and students due to issues such as uncertainty, the absence of unanimously established guidelines or the inequalities (or biases) that emerged during the state of emergency.

Reviewer 2 Report

10. The conclusion should include the research contribution

1. I suggest that this paragraph should be more concise. “Therefore, instead of traditional teaching, teachers have dealt an emergency remote teaching (ERT) defined as a transitory change from instructional delivery to an alternate mode of teaching due to crisis circumstances. It involves the use of fully remote teaching solutions for instruction or education that would otherwise be delivered face-to-face. Once the emergency has declined, didactic activities went back to the original format [6]. Consequently, teachers have experienced an important modification in their work format that it has been perceived as an abrupt and unplanned change since it has not been chosen by either learners or teachers [7].”

2. As the epidemic is a global problem. Therefore, there is no need to describe too much about the measures taken in Italy to prevent the epidemic.

3. Please add research hypotheses or research questions.

4. There is a problem with the small number of participants in this study. I think this is the biggest problem in the study.

5. Why not standardize the evaluation criteria for the questionnaire? This would make the article more difficult to read.6. Please present the validity results of the scale.

7. Please change section 2.3 to section 2.1.

8. Is this an experimental study? If not, what is the representativeness of the training participants?

9. The discussion is too general and should be written under subheadings according to the research hypothesis.

10. The conclusion should include the research contribution.

Round 2

Reviewer 2 Report

Dear Authors,

You have revised the manuscript and has improved significantly, so I will suggest editor that it can be accepted this revision.